# HARP: Hallucination Detection via Reasoning Subspace Projection

**Junjie Hu**     **Gang Tu**[*]     **ShengYu Cheng**     **Jinxin Li**     **Jinting Wang**

**Rui Chen**                **Zhilong Zhou**                **Dongbo Shan**

School of Computer Science and Technology
Huazhong University of Science and Technology
Wuhan, China
`{hujunjie,tugang}@hust.edu.cn`

## Abstract

Hallucinations in Large Language Models (LLMs) pose a major barrier to their reliable use in critical decision-making. Although existing hallucination detection methods have improved accuracy, they still struggle with disentangling semantic and reasoning information and maintaining robustness. To address these challenges, we propose **HARP** (**HA**llucination detection via **R**easoning subspace **P**rojection), a novel hallucination detection framework. HARP establishes that the hidden state space of LLMs can be decomposed into a direct sum of a semantic subspace and a reasoning subspace, where the former encodes linguistic expression and the latter captures internal reasoning processes. Moreover, we demonstrate that the Unembedding layer can disentangle these subspaces, and by applying Singular Value Decomposition (SVD) to its parameters, the basis vectors spanning the semantic and reasoning subspaces are obtained. Finally, HARP projects hidden states onto the basis vectors of the reasoning subspace, and the resulting projections are then used as input features for hallucination detection in LLMs. By using these projections, HARP reduces the dimension of the feature to approximately 5% of the original, filters out most noise, and achieves enhanced robustness. Experiments across multiple datasets show that HARP achieves state-of-the-art hallucination detection performance; in particular, it achieves an AUROC of 92.8% on TriviaQA, outperforming the previous best method by 7.5%.

## 1 Introduction

Large Language Models (LLMs) have recently demonstrated remarkable generative capabilities and broad applicability across various natural language processing tasks (Yang et al., 2024; Grattafiori et al., 2024; Minaee et al., 2024). However, hallucinations—i.e., model-generated information inconsistent with objective facts—remain a major obstacle to their deployment in critical decision-making scenarios (Ji et al., 2023; Huang et al., 2025). Consequently, efficiently and accurately detecting hallucinations during LLMs generation has become a pressing challenge.

From a cognitive perspective, the hallucination behavior of LLMs is to some extent similar to human's "nonsense" behavior. When answering complex questions, humans typically follow a "Reasoning → Expression" process: they first perform internal reasoning and then express part of the thought outcomes in language (Johnson-Laird, 1986). Therefore, although assessing the veracity of the answer is challenging when based solely on linguistic output, it can be substantially improved by observing the complete reasoning process (Frank & Goodman, 2012). By analogy, achieving high-precision hallucination detection in LLMs requires placing greater emphasis on the reasoning information encoded within the hidden states, rather than primarily on the semantic content of the outputs.

---

[*]Corresponding Author

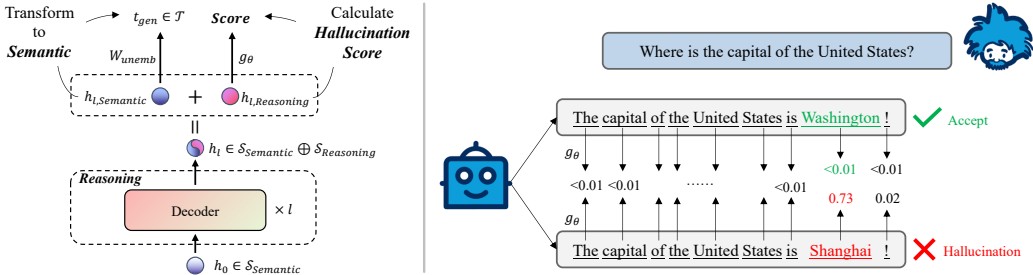

Figure 1: **Illustration of the proposed HARP framework for hallucination detection.** HARP separates the reasoning information $h_{l,\text{Reasoning}}$ from the hidden state $h_l$ to compute token-level hallucination scores, with the maximum score taken as the hallucination score of the entire response.

Inspired by this cognitive insight, we propose a novel hallucination detection framework, **HARP** (**HA**llucination detection via **R**easoning subspace **P**rojection). Specifically, HARP decomposes the hidden state space into a direct sum of the semantic subspace and the reasoning subspace. The semantic subspace captures the linguistic information of the generated content, while the reasoning subspace reveals the model's internal reasoning process. As illustrated in Figure 2, comparing humans and LLMs "Reasoning → Expression" behaviors reveals that LLMs discard reasoning information in the Unembedding layer while compressing semantic information into generated tokens. This suggests that the Unembedding layer inherently distinguishes be-

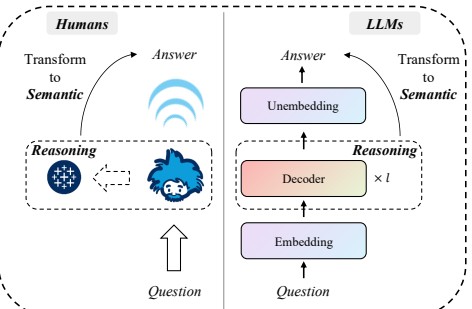

Figure 2: Comparison of the "Reasoning → Expression" behavior between humans and LLMs

tween semantic and reasoning information. Based on this, we perform Singular Value Decomposition (SVD) on the parameter matrix of the Unembedding layer to identify the basis vectors of the semantic subspace, which dominates token prediction, as well as those of the reasoning subspace, which is orthogonal to the semantic subspace.

Finally, HARP projects hidden states onto the basis vectors of the reasoning subspace and uses the resulting projections as input features for hallucination detection in LLMs. Since the reasoning subspace basis vectors account for only about 5% of the hidden state dimension, the input features are highly concentrated in reasoning information while largely eliminating noise. This allows HARP to achieve strong robustness while maintaining high detection accuracy. The main contributions of this work are:

- We establish that the hidden state space of LLMs can be decomposed into a direct sum structure composed of a semantic subspace and a reasoning subspace.

- We verify that the Unembedding layer has the capability to distinguish between the semantic subspace and the reasoning subspace. Furthermore, by performing SVD on the parameters of the Unembedding layer, the basis vectors that span the semantic subspace and the reasoning subspace are identified.

- We introduce a novel approach that explicitly constructs input features by projecting hidden states onto the basis vectors of the reasoning subspace. This projection drastically reduces the feature dimensionality to about 5% of the original, suppresses most noise, and achieves highly accurate hallucination detection in LLMs.

## 2 RELATED WORK

**Mechanistic interpretability of LLMs.** Research on mechanistic interpretability mainly focuses on two aspects: model parameters and hidden states. For the former, several works analyze weight matrices to uncover structural properties and interactions among modules. For instance, Merullo et al.

(2024) and Cheng et al. (2024) employ SVD to characterize attention-head structures and investigate their roles in downstream tasks. However, such approaches remain largely at the structural level, offering limited semantic interpretability. To address this, recent work has shifted toward analyzing hidden states directly, probing the predictive relationship between intermediate representations and model outputs (Gurnee et al., 2023; Lv et al., 2024; Ju et al., 2024; He et al., 2024; Jin et al., 2025).

**Hallucination detection.** The success of probing methods has motivated researchers to adopt similar ideas in hallucination detection (Marks & Tegmark, 2023; Bürger et al., 2024; Park et al., 2025). For instance, HaloScope (Du et al., 2024) leverages unlabeled embeddings and applies SVD to identify key subspace directions, followed by probing to link these directions to hallucinations. Yet, probing-based methods often rely on predefined supervised labels, making them less generalizable when feature dimensions are large or category priors are incomplete. Another line of work approaches hallucination detection from the perspective of output consistency. EigenScore (Chen et al., 2024) quantifies semantic agreement through covariance eigenvalues, while Farquhar et al. (2024) employ clustering and semantic entropy for hallucination detection. These methods are effective in practice but may suffer from misclassification due to their inability to exploit internal reasoning information.

Different from these approaches, our method explicitly separates semantic and reasoning subspaces, and projects hidden states onto the basis vectors of the reasoning subspace to construct compact and interpretable features for hallucination detection.

## 3 PRELIMINARIES

In this section, we first formulate a mathematical model to characterize the hallucination behavior of LLMs. Then, we analyze how the hidden state space evolves across decoder layers during generation, and subsequently decompose it into the direct sum of the semantic subspace and the reasoning subspace. This theoretical framework forms the foundation of HARP and provides essential support for hallucination detection via reasoning subspace projection.

### 3.1 MATHEMATICAL MODELING OF LLMS' HALLUCINATION

To model LLMs' hallucination mathematically, we first define the knowledge set known to the LLMs. Given an input sequence $x$ and its reference answer $y^*$, the LLMs generate multiple responses $\gamma = \{y^1, y^2, \ldots, y^s\}$ for $x$. If any generated response closely matches the reference answer, the knowledge about $x$ is considered known to the LLMs, denoted as $known(x) = 1$. Formally:

$$known(x) = \begin{cases} 1, & \exists y \in \gamma, sim(y, y^*) > \lambda \\ 0, & otherwise \end{cases} \tag{1}$$

where $sim(y, y^*)$ measures the similarity between $y$ and $y^*$, and $\lambda$ is a similarity threshold. Let $\mathcal{X}_{known} = \{x \mid known(x) = 1\}$ denote the set of all inputs whose knowledge is known to the LLMs. For each $x \in \mathcal{X}_{known}$, let $y = LLMs(x)$ denote the response generated by the LLMs. The hallucination indicator $G(y \mid x)$ is then defined as:

$$G(y \mid x) = \begin{cases} 1, & sim(y, y^*) \leq \lambda \\ 0, & otherwise \end{cases} \tag{2}$$

When $G(y \mid x) = 1$, the QA pair $[x, y]$ exhibits hallucination.

### 3.2 DIRECT SUM DECOMPOSITION OF HIDDEN STATE SPACE

Let the token vocabulary be $\mathcal{T}$. For LLMs with $l$ decoder layers, an input token $t \in \mathcal{T}$ is mapped by the embedding layer to an initial hidden state $h_0$ containing purely semantic information. As the hidden states propagate through successive layers, semantic and reasoning information are progressively integrated into their representations. Finally, the Unembedding layer projects only the semantic component to generate the output token $t_{gen} \in \mathcal{T}$. Thus, the final hidden state $h_l$ simultaneously encodes: (1) **Semantic prediction information**: To accurately generate the next token, $h_l$ must retain sufficient semantic features. These features are primarily captured by the parameter matrix $W_{unemb}$ of the Unembedding layer and play a dominant role in predicting the next token. (2)

**Reasoning trajectory information**: To support multi-step reasoning and intermediate state computation, $h_l$ also encodes intermediate reasoning information that does not directly affect the output. This information is typically not explicitly captured by $W_{unemb}$ and exerts minimal influence on the final output.

Denote the hidden state space at layer $l$ as $\mathcal{H}_l$. To disentangle these two signals, we decompose $\mathcal{H}_l$ into the direct sum of two orthogonal subspaces:

$$\mathcal{H}_l = \mathcal{S}_{Semantic} \oplus \mathcal{S}_{Reasoning} \tag{3}$$

where $\mathcal{S}_{Semantic}$ and $\mathcal{S}_{Reasoning}$ represent the semantic and reasoning subspaces, respectively. The final hidden state $h_l \in \mathcal{H}_l$ is projected to token logits by the Unembedding layer:

$$logits = W_{unemb} \cdot h_l \tag{4}$$

where $W_{unemb}$ denotes the Unembedding parameters. Let $h_{l,Semantic}$ and $h_{l,Reasoning}$ denote the components of $h_l$ in the semantic and reasoning subspaces, with $h_{l,Semantic}$ exerting primary influence on the $logits$ for token prediction, while $h_{l,Reasoning}$ encodes the model's reasoning processes.

To empirically validate the existence and functional role of the reasoning subspace, we design a *Reasoning Patch* experiment in Appendix E. This experiment demonstrates that the reasoning subspace $\mathcal{S}_{Reasoning}$ indeed captures critical intermediate reasoning information by showing that patching reasoning components from correct solutions can effectively rectify erroneous reasoning trajectories while preserving semantic coherence.

## 4 METHOD

In this section, we detail the proposed HARP framework for hallucination detection, as illustrated in Figure 1. First, in subsection 4.1, we validate the Unembedding layer's capability to effectively disentangle the semantic and reasoning subspaces. Then, in subsection 4.2 and subsection 4.3, we present a practical strategy for subspace decomposition. Finally, in subsection 4.4, we introduce the HARP algorithm, which performs hallucination detection based on reasoning subspace projection.

### 4.1 SUBSPACE DECOMPOSER — UNEMBEDDING LAYER

As shown in Figure 3, during token generation, the Unembedding layer of LLMs compresses only the semantic information $h_{l,Semantic}$ in hidden states into the generated tokens, filtering out the reasoning information $h_{l,Reasoning}$ used in intermediate computations. Therefore, by analyzing the basis vectors that interact with the Unembedding layer parameters $W_{unemb}$, we can determine the mathematical representations of the semantic subspace and its orthogonal reasoning subspace.

Based on the properties of the semantic and reasoning subspaces, their interactions with $W_{unemb}$ can be defined as:

$$W_{unemb} \cdot \mathcal{S}_{Semantic} \approx W_{unemb} \cdot \mathcal{H}_l \tag{5}$$

$$W_{unemb} \cdot \mathcal{S}_{Reasoning} \approx 0 \tag{6}$$

In other words, $\mathcal{S}_{Semantic}$ aligns with the principal acting directions of $W_{unemb}$, while the orthogonal $\mathcal{S}_{Reasoning}$ contributes negligibly to prediction scores. In subsection 5.3, we demonstrate the validity of our definitions for these subspace properties, laying the foundation for subsequently identifying the subspace basis vectors.

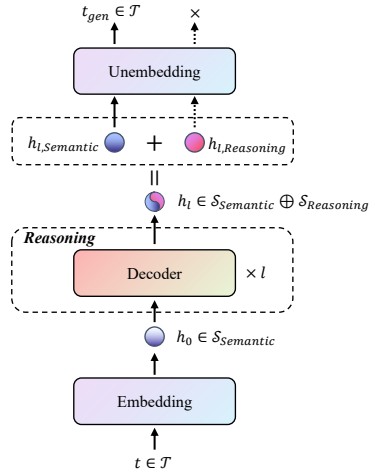

Figure 3: Flow of semantic and reasoning information within LLMs hidden states.

### 4.2 DETERMINATION OF SUBSPACE BASIS VECTORS VIA SVD

Given that the Unembedding layer can filter reasoning information, we first perform SVD on its parameter matrix $W_{unemb}$. By analyzing which hidden state components interact with $W_{unemb}$, we

identify the basis vectors of the semantic and reasoning subspaces. As shown in Equation 7, we decompose $W_{unemb}$ via SVD:

$$W_{unemb} = U\Sigma V^\top = \sum_{i=1}^{d} u_i \sigma_i v_i^\top \tag{7}$$

where $U \in \mathbb{R}^{\|\mathcal{T}\| \times \|\mathcal{T}\|}$, $\Sigma \in \mathbb{R}^{\|\mathcal{T}\| \times d}$, $V \in \mathbb{R}^{d \times d}$, $\|u_i\| = \|v_i\| = 1$, and the singular values in $\Sigma$ are sorted in descending order $\sigma_1 \geq \sigma_2 \geq \cdots \geq \sigma_k > \sigma_{k+1} = \sigma_{k+2} = \cdots = \sigma_d = 0$.

For any hidden state $h = \sum_{i=1}^{d} a_i v_i \in \mathbb{R}^d$, its interaction with $W_{unemb}$ is expressed as:

$$W_{unemb} \cdot h = \sum_{i=1}^{d} u_i \sigma_i v_i^\top \cdot a_i v_i = \sum_{i=1}^{d} (\sigma_i a_i) u_i \tag{8}$$

Since the vectors $u_i$ are mutually orthogonal, it follows that $W_{unemb} \cdot h = 0$ if and only if $\sum_{i=1}^{d} |\sigma_i a_i| = 0$, in which case the vector $h$ is filtered out by the Unembedding layer. In other words, $h$ belongs to the reasoning subspace $\mathcal{S}_{Reasoning}$ if and only if all singular values corresponding to non-zero $a_i$ vanish. Accordingly, we define an orthogonal basis for the reasoning subspace as $V_R = \{v_i \mid \sigma_i = 0\}$, while the remaining directions $V_S = \{v_i \mid \sigma_i > 0\}$ constitute the semantic subspace $\mathcal{S}_{Semantic}$. Since $\sigma_{i>k} = 0$, the semantic and reasoning subspaces can be expressed as:

$$\mathcal{S}_{Semantic} = \text{Span}\left(\{v_1, v_2, \ldots, v_k\}\right) \tag{9}$$

$$\mathcal{S}_{Reasoning} = \text{Span}\left(\{v_{k+1}, v_{k+2}, \ldots, v_d\}\right) \tag{10}$$

Let $a_i = v_i^\top h_l$ denote the projection coefficients of the hidden state $h_l$ onto the basis vectors. Then the components in the semantic and reasoning subspaces are $h_{l,Semantic} = \sum_{i=1}^{k} a_i v_i$ and $h_{l,Reasoning} = \sum_{i=k+1}^{d} a_i v_i$, respectively, with interactions with $W_{unemb}$ given by:

$$W_{unemb} \cdot h_{l,Semantic} = \sum_{i=1}^{k} \sigma_i(a_i u_i) = W_{unemb} \cdot h_l \tag{11}$$

$$W_{unemb} \cdot h_{l,Reasoning} = \sum_{i=k+1}^{d} \sigma_i(a_i u_i) = 0 \tag{12}$$

This partitioning of the hidden state space aligns precisely with the definitions of semantic and reasoning subspaces in Equation 5 and Equation 6, and provides a theoretical basis for constructing low-rank approximation-based subspaces in real models.

### 4.3 CONSTRUCTION OF SEMANTIC AND REASONING SUBSPACES VIA LOW-RANK APPROXIMATION

While the method described in subsection 4.2 can ideally construct the semantic and reasoning subspaces, in practice, the condition $\sigma = 0$ for singular values rarely holds. To address this, we perform a rank-$k$ approximation of $W_{unemb}$, extracting the $k$ most representative semantic directions from its row space to define the semantic subspace under realistic conditions, and determine the reasoning subspace using orthogonal relationships.

Specifically, based on Equation 7, for any $k < rank(W_{unemb})$, the Eckart–Young–Mirsky theorem (Eckart & Young, 1936; Greenacre et al., 2022) gives the best rank-$k$ approximation $W_k$ of $W_{unemb}$ under the Frobenius norm as:

$$W_k = \underset{\text{rank}(A) \leq k}{\arg\min} \|W_{unemb} - A\|_F = \sum_{i=1}^{k} u_i \sigma_i v_i^\top \tag{13}$$

To ensure that this approximation does not significantly degrade prediction accuracy, the following information-preservation condition should hold:

$$\|W_{unemb} - W_k\|_F = \sqrt{\sum_{i=k+1}^{d} \sigma_i^2} \ll \sqrt{\sum_{i=1}^{k} \sigma_i^2} \tag{14}$$

This condition implies that $W_k$ retains the majority of $W_{unemb}$'s information in the Frobenius norm, i.e., the first $k$ singular values account for most of the total energy.

Figure 4a illustrates the singular value distribution of the Unembedding layer parameters. We observe that the trailing 5% of singular values are markedly smaller than the others, and the information loss associated with these minor singular values can be safely ignored. Accordingly, we

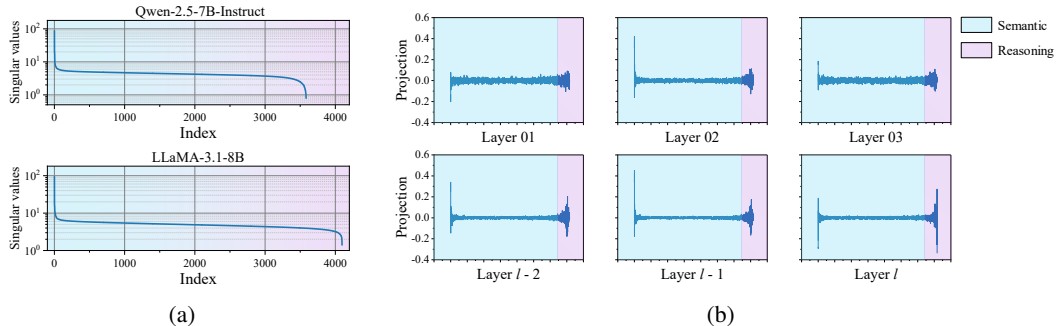

(a)                                                   (b)

Figure 4: (a) Singular value distributions of $W_{unemb}$ after SVD, with hidden state dimensions of 3584 for Qwen-2.5-7B-Instruct and 4096 for LLaMA-3.1-8B. (b) Projections of hidden states onto the basis vectors of the semantic and reasoning subspaces across layers, where the first row shows the first three layers and the second row shows the last three layers. Further details are provided in Appendix B.

set $k = d \times 95\%$. By analyzing $W_k$ and incorporating it into Equation 9 and Equation 10, we derive the corresponding subspace representations. Denoting the basis of the reasoning subspace as $V_R = [v_{k+1}, v_{k+2}, \ldots, v_d] \in \mathbb{R}^{d \times (d-k)}$, the projection of hidden states $h_l$ onto the reasoning subspace is:

$$proj_R(h_l) = V_R^\top \cdot h_l \tag{15}$$

In subsection 5.3, we experimentally demonstrate that replacing $W_{unemb}$ with $W_k$ in the token prediction task introduces negligible error. This finding provides the basis for subsequently using $proj_R(h_l)$ as the input feature to construct the hallucination detector.

### 4.4 HALLUCINATION DETECTION VIA REASONING SUBSPACE PROJECTION

As shown in Figure 4b, universal representations of hidden states are extracted from different layers of the LLMs and projected onto the basis $V = [V_S, V_R]$. We observe that shallow hidden states primarily enhance information in the semantic subspace, while deep hidden states exhibit stronger features in the reasoning subspace. This observation is consistent with our definitions of the two subspaces. Based on this, we propose a novel hallucination detection framework—HARP, which detects hallucinations using projections of hidden states onto the reasoning subspace.

During training, HARP employs a beam search strategy to generate multiple candidate answers $\gamma = \{y^1, y^2, \ldots, y^s\}$ for a given question $x$, and annotates whether each candidate contains hallucinations. For a QA pair $[x, y]$ composed of $n$ tokens, HARP computes the projection of each token's hidden state onto the reasoning subspace and calculates its hallucination score. The maximum score among all tokens is taken as the hallucination score of the QA pair:

$$g_\theta(y|x) = \max_{1 \le i \le n} g_\theta\left(proj_R\left(h_l^{(i)}\right)\right) \tag{16}$$

where $\theta$ denotes the parameters of the hallucination detector. $g_\theta\left(proj_R\left(h_l^{(i)}\right)\right)$ represents the hallucination score of the $i$-th token, and $g_\theta(y|x) \in [0, 1]$ is the score for the entire QA pair. We optimize the detector using the Binary Cross-Entropy Loss (Goodfellow et al., 2016):

$$\mathcal{L} = -flag \cdot \log(g_\theta) - (1 - flag) \cdot \log(1 - g_\theta) \tag{17}$$

where $flag \in \{0, 1\}$ indicates whether the QA pair $[x, y]$ contains hallucinations. Minimizing this loss trains a hallucination detector $\widehat{G}$:

$$\widehat{G}(y|x) = \mathbb{I}\left[g_\theta(y|x) > \alpha\right] \tag{18}$$

where $\alpha \in [0, 1]$ is the detection threshold. When $\widehat{G}(y|x) = 1$, the QA pair is considered hallucinated. Beam search is used only during training to construct diverse supervision samples, whereas during testing, $\widehat{G}$ relies solely on the projection of a single sampled answer onto $\mathcal{S}_{Reasoning}$.

As shown in Figure 1, for the question "*Where is the capital of the United States?*", the hallucinated answer "*The capital of the United States is Shanghai!*" assigns a hallucination score of 0.73 to the token "*Shanghai*", whereas all tokens in the correct answer "*The capital of the United States is Washington!*" have scores below 0.01. This demonstrates the effectiveness of $\widehat{G}$.

## 5 EXPERIMENTS

In this section, we first describe the experimental setup and demonstrate HARP's advantages over other hallucination detection methods across multiple models and datasets. We then analyze the validity of our proposed direct-sum decomposition of the hidden state space and the necessity of the projection operation, followed by an evaluation of the detection performance under varying reasoning subspace dimensions and hallucination score thresholds. Finally, we discuss HARP's cross-dataset generalization capability.

### 5.1 EXPERIMENTAL SETUP

**Datasets and models.** Our experiments cover four generative question answering (QA) tasks, including three open-domain dialogue QA datasets—NQ Open (Kwiatkowski et al., 2019), TruthfulQA (Lin et al., 2022a) (generation task), and TriviaQA (Joshi et al., 2017)—and one reading comprehension dataset, TyDiQA-GP (English) (Clark et al., 2020). To assess the effectiveness and generality of our proposed framework, we conduct evaluations using two widely adopted open-source foundation models: Qwen-2.5-7B-Instruct (Yang et al., 2024) and LLaMA-3.1-8B (Grattafiori et al., 2024). More dataset and inference details are provided in Appendix A.

**Evaluation Metrics.** AUROC (area under the ROC curve) is employed as the primary evaluation metric. AUROC measures a binary classifier's ability to distinguish positive and negative samples across different thresholds, ranging from 0 to 1, with higher values indicating stronger discriminative power. AUROC equal to 1 indicates perfect classification, while a value of 0.5 corresponds to random guessing.

**Baseline Methods.** HARP is compared with several mainstream hallucination detection methods, including Perplexity (Ren et al., 2023), LN-Entropy (Malinin & Gales, 2021), Semantic Entropy (Farquhar et al., 2024), Lexical Similarity (Lin et al., 2022b), EigenScore (Chen et al., 2024), and HaloScope (Du et al., 2024).

**Correctness Measurement.** Following Chen et al. (2024), correctness is determined based on ROUGE-L and semantic similarity between generated and reference answers. Semantic similarity is computed using the BLEURT model (Sellam et al., 2020; Park et al., 2025). An answer is considered correct if its ROUGE-L score exceeds 0.7 or its semantic similarity exceeds 0.5.

### 5.2 MAIN RESULTS

Table 1 summarizes the AUROC scores (in %) of various hallucination detection methods across four QA datasets, using Qwen-2.5-7B-Instruct and LLaMA-3.1-8B as backbone models. Several key findings emerge from these results. (1) HARP consistently outperforms all baseline methods across all datasets and models, often by a significant margin. For instance, on TriviaQA, HARP achieves AUROC scores of 92.8% on Qwen and 92.9% on LLaMA, yielding improvements of +7.5% and +16.6%, respectively, over the second-best method, demonstrating its robustness and scalability across architectures and data characteristics. (2) Baseline methods such as Perplexity and HaloScope perform competitively on simpler datasets like TriviaQA, where answers are often limited to one or two tokens, but their performance deteriorates sharply on more complex datasets such as TyDiQA, which contains long contexts and accompanying documents. In contrast, HARP maintains high AUROC scores of 88.4% on Qwen and 86.6% on LLaMA in these challenging settings, highlighting its ability to handle reasoning-intensive and context-rich inputs. (3) Sampling-based methods, such as Semantic Entropy, Lexical Similarity, and EigenScore, incur higher computational costs but still fail to achieve comparable performance, whereas HARP's single-pass approach provides both superior efficiency and accuracy.

In addition, Table 2 reports the number of known and unknown questions for Qwen-2.5-7B-Instruct across the four datasets, reflecting the model's varying answering capabilities on these benchmarks.

Table 1: **Main result.** Comparison of different methods on hallucination detection performance across multiple datasets. All values are AUROC percentages. "Single" indicates whether multiple samplings are required for hallucination detection.

| Models | Methods | Single | NQ Open | TruthfulQA | TriviaQA | TyDiQA |
|---|---|---|---|---|---|---|
| Qwen-2.5-7B-Instruct | Perplexity | ✓ | 76.5 | 64.4 | 83.1 | 30.5 |
| | LN-Entropy | ✗ | 77.7 | 63.6 | 80.2 | 47.1 |
| | Semantic Entropy | ✗ | 77.7 | 60.0 | 76.1 | 68.6 |
| | Lexical Similarity | ✗ | 77.8 | 63.9 | 76.9 | 60.3 |
| | EigenScore | ✗ | 78.9 | 63.8 | 76.2 | 74.8 |
| | HaloScope | ✓ | 60.7 | 63.0 | 85.3 | 69.0 |
| | **HARP(Ours)** | ✓ | **84.0** | **88.1** | **92.8** | **88.4** |
| LLaMA-3.1-8B | Perplexity | ✓ | 50.3 | 71.4 | 76.3 | 53.4 |
| | LN-Entropy | ✗ | 52.7 | 62.5 | 55.8 | 48.8 |
| | Semantic Entropy | ✗ | 60.7 | 59.4 | 68.7 | 62.2 |
| | Lexical Similarity | ✗ | 60.9 | 49.1 | 71.0 | 69.5 |
| | EigenScore | ✗ | 56.7 | 45.3 | 69.1 | 82.4 |
| | HaloScope | ✓ | 62.7 | 70.6 | 76.2 | 53.3 |
| | **HARP(Ours)** | ✓ | **89.4** | **88.5** | **92.9** | **86.6** |

Collectively, these findings validate the effectiveness, robustness, and practical utility of HARP for hallucination detection in diverse QA scenarios.

Table 2: **Distribution of known and unknown questions across four QA datasets.** A question is classified as **Known** if the model's knowledge state contains the correct answer according to the criterion in Equation 1, and as **Unknown** if none of the 10 candidate responses contain the correct answer.

| Dataset | Known | Unknown |
|---|---|---|
| TruthfulQA | 636 | 181 |
| TyDiQA | 402 | 38 |
| TriviaQA | 6225 | 3735 |
| NQ-open | 293 | 3317 |

## 5.3 More Analysis

**Rationality of Direct Sum Decomposition in Hidden State Space.** To validate this direct sum decomposition, we conduct a comparative experiment: removing the reasoning subspace components of hidden states and examining their effect on token prediction scores and rankings. Mathematically, this operation can be formulated as:

$$logits' = W_k \cdot h_l = W_{unemb} \cdot h_{l,Semantic} \qquad (19)$$

As shown in Figure 5a, computing token prediction scores using Equation 19 instead of the original $logits$ maintains the top rankings of greedily generated tokens. This result aligns with our theoretical design: the hidden state space can be decomposed into semantic and reasoning subspaces, and token prediction is mainly influenced by the semantic subspace component $h_{l,Semantic}$. This experiment confirms that the proposed direct sum decomposition exhibits clear representational disentanglement and functional partitioning, providing theoretical support for building hallucination detection models based on the reasoning subspace.

**Ablation Study.** We tested the importance of projecting hidden states onto the reasoning subspace by comparing hallucination detection performance under different projection strategies. "HARP (w/o)" denotes completely removing the projection, while retaining hidden state features of the same dimensionality as full HARP; "HARP (random)" denotes randomly selecting a set of bases from the projection basis $V = \{v_1, v_2, \ldots, v_d\}$ for projection. The results in Table 3 show that both removing the projection and using random projection significantly degrade hallucination detection performance, confirming the necessity of projecting hidden states onto the reasoning subspace.

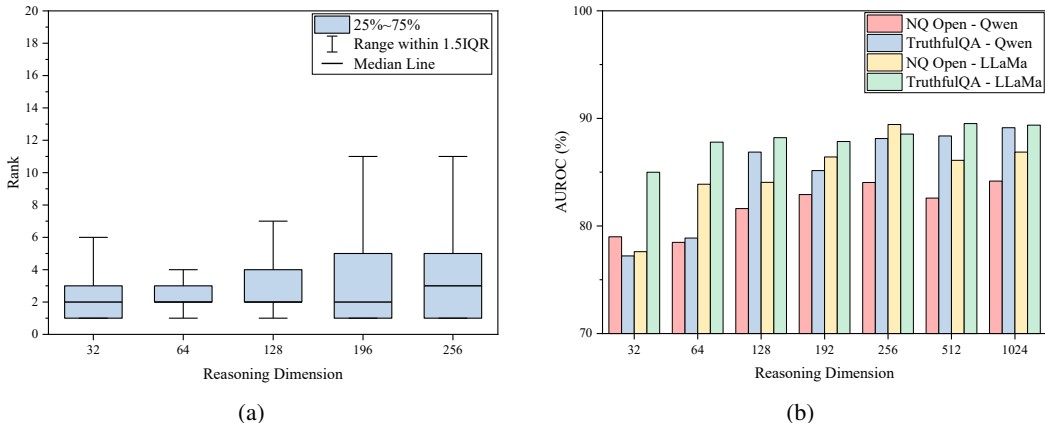

Figure 5: (a) Greedy token rankings in $logits'$ under different reasoning subspace dimensions. (b) Effect of reasoning subspace dimension on hallucination detection performance.

Table 3: Hallucination detection performance under different projection strategies

| Methods | Qwen-2.5-7B-Instruct | | LLaMA-3.1-8B | |
|---|---|---|---|---|
| | NQ Open | TruthfulQA | NQ Open | TruthfulQA |
| HARP (w/o) | 62.9 | 70.7 | 70.4 | 73.5 |
| HARP (random) | 67.6 | 68.6 | 59.5 | 75.8 |
| **HARP** | **84.0** | **88.1** | **89.4** | **88.5** |

**Impact of Reasoning Subspace Dimension on Hallucination Detection.** The reasoning subspace dimension affects hallucination detection in two ways: (1) its influence on $logits$ scores: when the dimension is too large, Equation 14 gradually breaks down, which impairs the model's next-token prediction capability; (2) its effect on detection accuracy and generalization: increasing the dimension may improve training accuracy but also increases the risk of overfitting, reducing generalization. We evaluated dimensions from 32 to 1024 using Qwen-2.5-7B-Instruct and LLaMA-3.1-8B models. As shown in the Figure 5b, a dimension of 256 yields the best performance. This dimension accounts for only about 5% of the original hidden state dimensionality, preserving sufficient reasoning information while filtering most redundant noise, satisfying the information-preservation constraint in Equation 14.

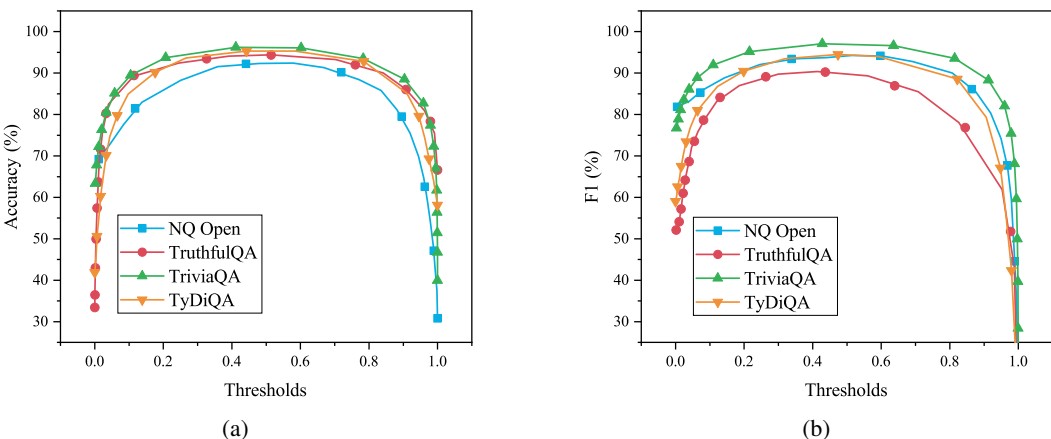

Figure 6: (a) Effect of hallucination score threshold on detection accuracy. (b) Effect of hallucination score threshold on detection F1 score.

**Selection of Hallucination Score Threshold.** In practice, it is necessary to set a hallucination score threshold $\alpha$ so that $\widehat{G}(y|x)$ produces a clear binary decision. As shown in Figure 6a and Figure 6b, when $\alpha$ is between 0.2 and 0.8, both detection accuracy and F1 score remain high, indicating a substantial separation between normal and hallucinated answers under $\widehat{G}$. To align with common expectations for a binary classifier, we set $\alpha = 0.5$, where $\widehat{G}(y|x) = \mathbb{I}\left[g_\theta(y|x) > 0.5\right]$.

**Robustness Experiments.** To apply HARP in real-world scenarios, we examined its performance under distribution shifts between training and test sets. We trained the hallucination detector on a source dataset $s$ and evaluated it on different target datasets $t$. Figure 7 shows that HARP generalizes well across multiple target datasets. Notably, when trained on TriviaQA, its accuracy on NQ Open is nearly identical to directly training on NQ Open, demonstrating HARP's strong robustness and cross-distribution adaptability.

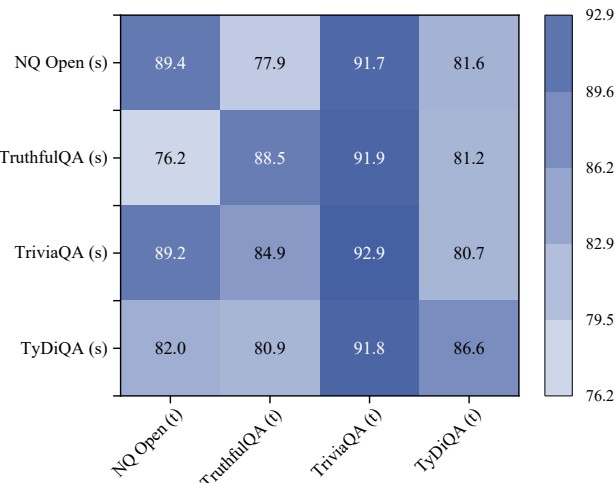

Figure 7: Cross-dataset generalization. "(s)" indicates the source dataset used for training the hallucination detector; "(t)" indicates the target dataset.

## 6 CONCLUSION

In this study, we introduced HARP, a novel hallucination detection method that leverages only reasoning information as input features, achieving high detection accuracy while maintaining strong robustness. First, we showed that the hidden state space admits a direct-sum decomposition into a semantic subspace and a reasoning subspace, and that the Unembedding layer can effectively separate these two components. Building on this, we applied singular value decomposition to the parameters of the Unembedding layer and, following the Eckart–Young–Mirsky theorem, approximated $W_{unemb}$ with its best rank-$k$ approximation $W_k$. Setting $k = d \times 95\%$, we identified basis vectors for both the semantic and reasoning subspaces that align with empirical observations. Furthermore, we empirically validated that the reasoning subspace effectively captures intermediate reasoning information through the Reasoning Patch experiment detailed in Appendix E. Finally, HARP constructs an accurate and efficient hallucination detector by using the projections of hidden states in the reasoning subspace as input features. Experiments show that HARP significantly outperforms existing mainstream hallucination detection methods and maintains robustness under distribution shifts across datasets. In addition, we present a proof-of-concept demonstration of hallucination mitigation using our framework in Appendix D and aim to inspire future research in this direction. The authors acknowledge OpenAI's ChatGPT (OpenAI, 2025) for its support in language polishing of this manuscript.

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

APPENDIX

# A  DATASETS AND IMPLEMENTATION DETAILS

**Input prompts.** In our experiments, datasets were categorized based on whether additional supporting information is provided. For datasets without context, including NQ-Open, TruthfulQA, and TriviaQA, we used prompts that contain only the question. Specifically, the prompt format is:

---
**Prompts for datasets without context**

Q: {question}
A:

---

For datasets with context, including TyDiQA, the prompt includes both the task description and the relevant context:

---
**Prompts for datasets with context**

Concisely answer the following question based on the information in the given passage:
Passage: {context}
Q: {question}
A:

---

**Implementation details.** Using the formulations in subsection 3.1, we select LLMs' known knowledge set $\mathcal{X}_{known} = \{x \mid known(x) = 1\}$ and unknown knowledge set $\mathcal{X}_{unknown} = \{x \mid known(x) = 0\}$. 75% of $\mathcal{X}_{known}$ is used for training, while the remaining 25%, together with $\mathcal{X}_{unknown}$, is used to test the hallucination detector on unseen data. For dataset questions, the temperature is set to 0.5, and beam search is used to generate 10 answer paths per question. The hallucination detector $G$ is a two-layer MLP with hidden dimension 1024 and ReLU activation. Training is conducted for 50 epochs with the Adam optimizer, initial learning rate 1e-4, cosine decay, batch size 128, and weight decay 3e-4.

# B  EXTRACTING A UNIVERSAL REPRESENTATION VIA UNCENTERED PCA

Given a collection of $n$ hidden vectors $\{h^{(i)}\}_{i=1}^n$ from LLMs, each of dimension $d$, we arrange them into a matrix:

$$M = \begin{bmatrix} (h^{(1)})^\top \\ \vdots \\ (h^{(n)})^\top \end{bmatrix} \in \mathbb{R}^{n \times d} \tag{20}$$

From an energy-maximization perspective, the "universal representation" of these hidden vectors can be interpreted as their dominant direction of variation in the feature space. To extract this direction, we perform SVD:

$$M = U'\Sigma'V'^\top \tag{21}$$

where $U' \in \mathbb{R}^{n \times n}$, $\Sigma' = \text{diag}(\sigma'_1, \cdots, \sigma'_d) \in \mathbb{R}^{n \times d}$, $V'^\top = [v'_1, \cdots, v'_d] \in \mathbb{R}^{d \times d}$, and the singular values satisfy $\sigma'_1 \geq \sigma'_2 \geq \cdots \geq 0$. The dominant right singular vector $v_1$ provides the principal direction of the row space of $M$, which is equivalent to the first principal component in uncentered Principal Component Analysis (PCA). We define the *universal representation direction* as:

$$\hat{h} = v'_1 \in \mathbb{R}^d \tag{22}$$

By collecting $n$ hidden states from the $i$-th layer, we can derive the corresponding universal representation $\hat{h}_i$ following the steps above. Projecting it onto the basis vectors $V = [V_S, V_R] \in \mathbb{R}^{d \times d}$ yields the projections of the $i$-th layer's hidden state onto the semantic and reasoning subspaces:

$$proj\left(\hat{h}_i\right) = V^\top \cdot \hat{h}_i \tag{23}$$

In Figure 4b, we normalize the lengths of $proj\left(\hat{h}_i\right)$ and visualize the projections of the universal representations of hidden states from the first three and last three layers of the Qwen-2.5-7B-Instruct model onto the semantic and reasoning subspaces. We observe that shallow layer vectors are primarily represented in the semantic subspace, while deep layer vectors are more concentrated in the reasoning subspace.

## C   ANALYSIS OF LAYER-WISE CONTRIBUTIONS IN LLMS

Although our previous analysis has characterized the hidden states after processing through multiple decoder layers, it remains important to understand the individual contributions of each layer and how they differ. To this end, we define the contribution of the $i$-th decoder layer as $dh_i = h_i - h_{i-1}$, and, following the method described in Appendix B, compute the universal representation direction $\hat{dh}_i$. Since singular vectors obtained via SVD can have arbitrary signs, we compute the absolute cosine similarity between $\hat{dh}_i$ and $\hat{dh}_j$ to measure the similarity between the universal representations of the increments of the layers $i$ and $j$.

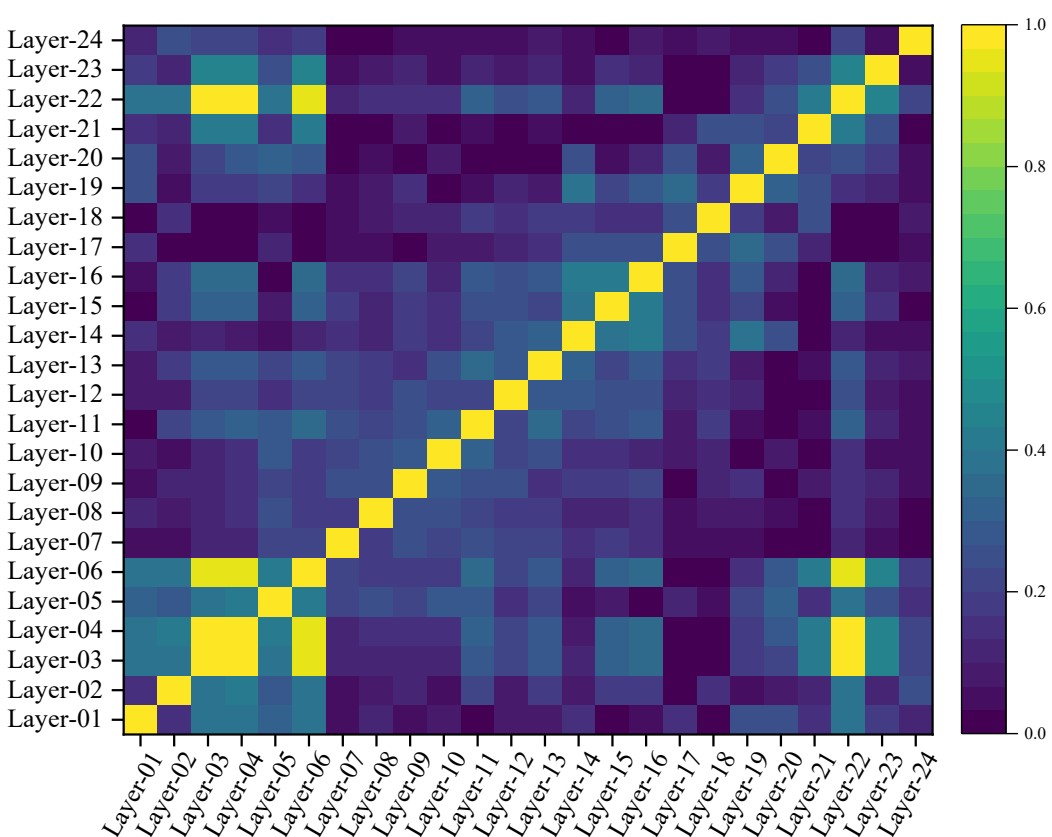

Figure 8: Similarity between universal representation directions of layer-wise increments

Figure 8 illustrates the cosine similarity between the universal representation directions of layer-wise increments in the Qwen-2.5-0.5B-Instruct model. We observe that the first six layers behave in a broadly similar manner; however, the first two layers are relatively independent of the remaining ones, while layers 3, 4, and 6 exhibit almost identical directions. Interestingly, the direction around layer 22 is remarkably similar to that of layers 3, 4, and 6. We hypothesize that the first two layers primarily parse the shallow semantic structure of the input, layers 3, 4, and 6 encode this information into an internal representation space (a "language" specific to the LLM), the intermediate layers perform various reasoning operations over this representation, and layer 22 decodes it back into a human-interpretable semantic space before unembedding.

Based on this observation, we argue that mitigating hallucinations—especially those induced by suboptimal training patterns or aggressive answer-generation strategies—requires intervention in the decoding behavior around layer 22. Following this intuition, Appendix D provides a demonstration of hallucination mitigation, with the goal of inspiring future research in this direction.

# D MITIGATING HALLUCINATIONS BY MODIFYING REASONING SUBSPACE COMPONENTS

Based on our previous analysis of LLM behavior, we explore mitigating hallucinations by intervening on the components of hidden states within the reasoning subspace. To this end, we introduce a fictitious city, "*Epsilon*", and pose the question to the LLM: "*The capital of Epsilon is ?*". We then investigate the effect of removing the reasoning subspace components from hidden states at different layers and observe the resulting impact on the LLM's outputs.

Table 4: Impact of interventions across layers and reasoning subspace dimensions on hallucination mitigation for the query. Green responses indicate a correct mitigation of hallucinations.

| Intervened Layer | Reasoning Subspace Dimension | Output |
|---|---|---|
| None | None | Epsilon's capital is likely **Kaiyuan**, which is the capital city of the Kingdom of Kaiyuan in the Eastern Regions. |
| Layer-01 | 8 | Epsilon's capital is likely to be the city or town where its government and administrative center is located. |
| | 16 | Epsilon's capital is likely to be the city or town where its government and administrative center is located. |
| | 32 | Epsilon's capital is likely to be the city or town where its government and administrative center is located. |
| | 64 | The capital of Epsilon is Elea. |
| Layer-02 | 8 | Epsilon is the capital city of the planet Alpha. |
| | 16 | I apologize, but I don't have enough context to determine the specific name or location of the capital city in question. |
| | 32 | I apologize, but I'm not able to determine the capital city of Epsilon as it appears to be a fictional planet or alternate universe. |
| | 64 | Epsilon is the capital city of the European Union (EU). |
| Layer-22 | 8 | I'm sorry, but I don't have enough context to accurately answer your question about the capital city of Epsilon. |
| | 16 | I'm sorry, but I need more context to accurately answer your question. |
| | 32 | I'm sorry for any misunderstanding earlier. |
| | 64 | Epsilon is currently not specified in my knowledge base for now. |
| Layer-23 | 8 | I'm sorry, but I don't have enough context to accurately answer your question about the capital city of Epsilon. |
| | 16 | I'm sorry, but I need more information to accurately answer your question. |
| | 32 | Epsilon is currently not in my knowledge base as I am an AI language model created by Alibaba Cloud based on publicly available information... |
| | 64 | Epsilon is currently unknown due to lack of information about its current status in relation to other planets in our solar system or neighboring celestial bodies... |

Table 4 presents the outputs of the LLM under interventions in various layers and with different subspace dimensions of reasoning. We observe that interventions in shallow layers, such as layers 1 and 2, produce limited improvement, whereas interventions at deeper layers, such as layers 22 and 23, lead the LLM to explicitly acknowledge its lack of knowledge about the fictitious city "*Epsilon*"

and refuse to answer. This phenomenon aligns with our earlier analysis of the behavior of LLMs. We hope that this hallucination-mitigation demo can inspire further research in this direction.

# E    VERIFICATION OF REASONING INFORMATION IN THE REASONING SUBSPACE

To verify that the components of hidden states lying in the reasoning subspace indeed encode internal reasoning information, we design a controlled experiment consisting of three input conditions (Figure 9). These conditions isolate the effect of the reasoning subspace while keeping all other factors unchanged.

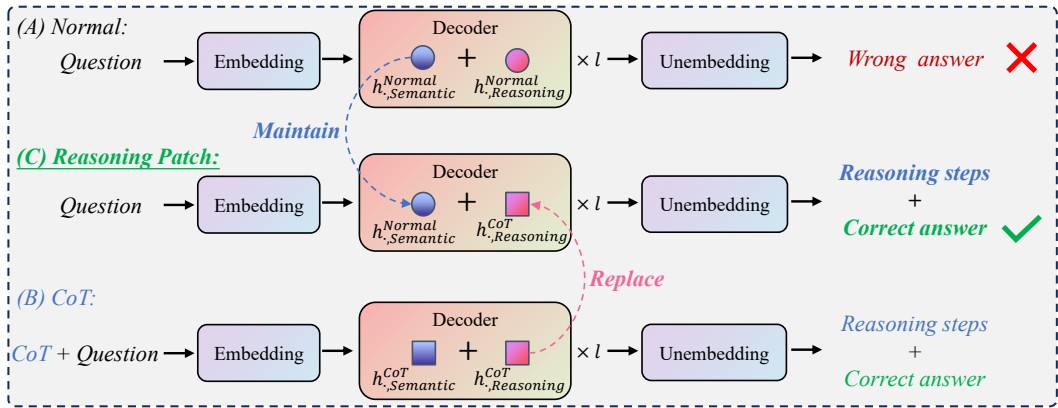

Figure 9: **Experimental design illustrating the three conditions used to verify the causal role of the reasoning subspace.** *(A)Normal*: the model receives only the question and produces an incorrect answer. *(B)CoT*: a chain-of-thought is prepended, enabling multi-step reasoning and a correct answer. *(C)Reasoning Patch*: no CoT is provided, but the reasoning-subspace components of hidden states at all layers are replaced with those from the CoT run, causing the model to generate reasoning steps and arrive at the correct answer.

**(A) Normal: direct question input.**    In the first condition, we feed the model only the question without any chain-of-thought (CoT) guidance. The model typically produces an incorrect answer. Let the hidden state be

$$h_{\cdot}^{Normal} = h_{\cdot,Semantic}^{Normal} + h_{\cdot,Reasoning}^{Normal}.$$

**(B) CoT: prepend chain-of-thought.**    In the second condition, we prepend a chain-of-thought (Wei et al., 2022) to the input. The model now first generates intermediate reasoning steps and then outputs the correct answer. The hidden state is

$$h_{\cdot}^{CoT} = h_{\cdot,Semantic}^{CoT} + h_{\cdot,Reasoning}^{CoT}.$$

**(C) Reasoning Patch: replace reasoning components at all relevant layers.**    The third condition serves as the key causal intervention. The input text is identical to condition (A); however, at every decoder layer that contributes to the representation of a token, we replace only the reasoning-subspace component of the hidden state with the corresponding component extracted from condition (B). Formally, for all layers along the forward-pass trajectory of token $t$, we apply:

$$h_{\cdot}^{Patch} = h_{\cdot,Semantic}^{Normal} + h_{\cdot,Reasoning}^{CoT}.$$

Thus, semantic information is preserved at every layer, while the reasoning components across all intermediate layers are substituted with those from the CoT run. This ensures that the patched forward pass follows the CoT reasoning trajectory throughout the entire decoder stack.

**Key result.**    We evaluate the effectiveness of the proposed Reasoning Patch on mathematical reasoning benchmarks such as GSM8K (Cobbe et al., 2021), using both *few-shot CoT* and *zero-shot CoT* to extract the reasoning-subspace components.

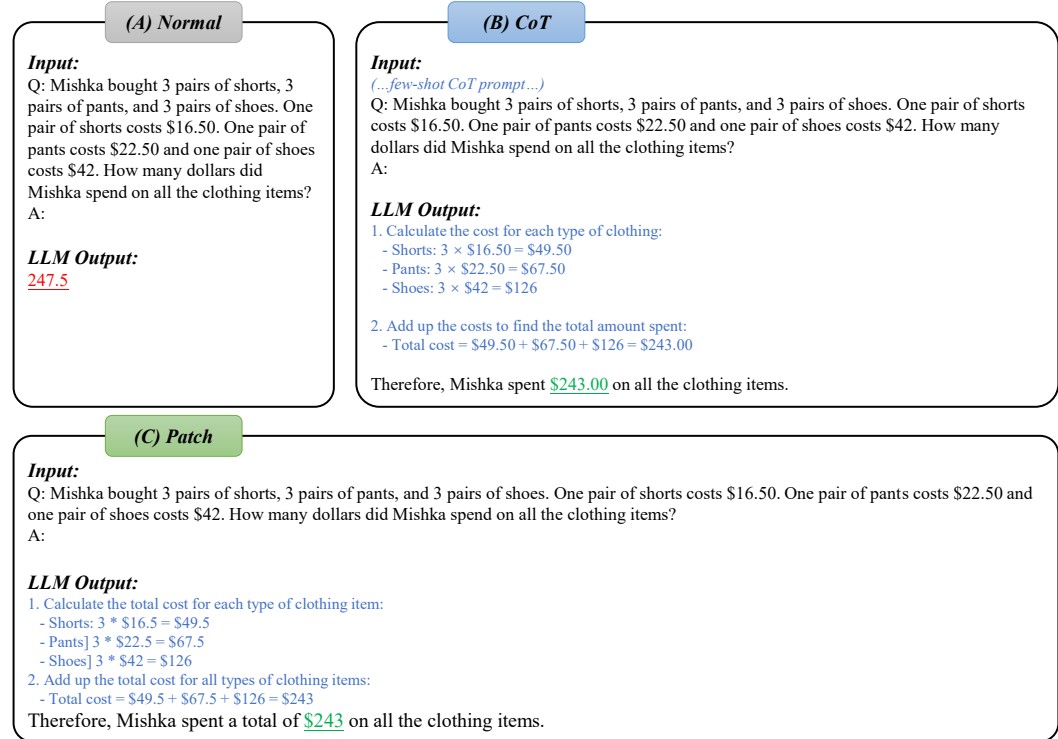

Figure 10: Reasoning Patch experiment using *few-shot* chain-of-thought supervision.

Figure 10 presents the Qwen2.5-7B-Instruct outputs under the three conditions (A)–(C) when the reasoning components of condition (C) are derived from *few-shot CoT*, with the full prompts shown in Table 5. We observe that, even though condition (C) receives *no* CoT text in the input, injecting the CoT-derived reasoning-subspace components reliably triggers the model to follow a "reason-then-answer" generation pattern. As a result, the model transitions from an incorrect answer in (A) to a correct, multi-step reasoning process in (C), demonstrating that the patched reasoning trajectory causally determines the emergence of correct step-by-step reasoning.

Table 5: *few-shot* chain-of-thought prompt.

---

Q: A robe takes 2 bolts of blue fiber and half that much white fiber. How many bolts in total does it take?
A: **A robe needs 2 bolts of blue fiber.**
**The amount of white fiber needed is half of the blue fiber.**
**Half of 2 bolts is 1 bolt of white fiber.**
**The total bolts needed is the sum of blue and white fiber.**
**2 bolts plus 1 bolt equals 3 bolts.**
Therefore, the final answer is 3.

(…***Input***…)

---

Figure 11 shows the corresponding results when the reasoning components are extracted from *zero-shot CoT*. Remarkably, even though condition (C) does not contain the zero-shot instruction (e.g., "Answer the following question step by step to the best of your ability."), the patched model nonetheless produces a coherent step-by-step reasoning chain before giving the final answer. Interestingly, in this setting the original CoT run in condition (B) makes an arithmetic mistake and outputs an incorrect final answer; however, condition (C)—which inherits only the reasoning-subspace components rather than the explicit token sequence—does *not* reproduce this error and instead produces the cor-

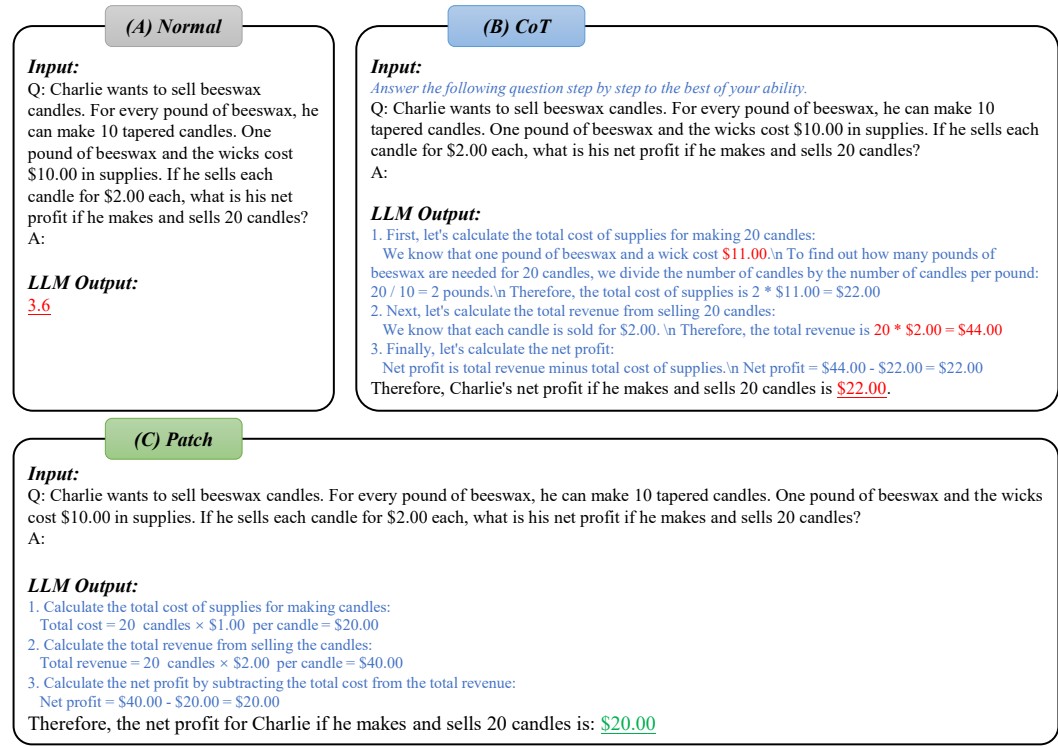

Figure 11: Reasoning Patch experiment using *zero-shot* chain-of-thought prompting.

rect result. This highlights that the reasoning subspace captures the structural reasoning trajectory without being constrained by the semantic information in the CoT prompt.

Together, these results provide compelling evidence that the reasoning subspace encodes causally meaningful internal reasoning information, and that injecting its components is sufficient to induce coherent multi-step reasoning even in the absence of explicit CoT prompting.

## F  COMPUTATIONAL COMPLEXITY OF SVD

To construct the reasoning subspace, we perform singular value decomposition (SVD) on a matrix $M \in \mathbb{R}^{n \times d}$, where $n$ denotes the vocabulary size and $d$ is the dimensionality of the hidden representation. In typical large language models, the matrix is tall and skinny with $n \gg d$ (e.g., for Qwen2.5-7B, $n = 152,064$ and $d = 3,584$). The computational complexity of SVD depends on these matrix dimensions as well as whether a full or truncated decomposition is applied.

**Time Complexity.** For a full SVD on an $n \times d$ matrix, the time complexity is

$$O\big(\min(nd^2,\, n^2d)\big).$$

Since the vocabulary size is typically much larger than the hidden dimension, the dominant term becomes

$$O(nd^2),$$

which makes full SVD computationally expensive in practice. For truncated SVD that retains only the top-$k$ singular directions, the complexity reduces to

$$O(ndk),$$

particularly when using iterative or randomized SVD algorithms. Such approximations are crucial for scaling to vocabularies of realistic size.

Table 6: SVD computation cost on the unembedding layer using an H100 80GB GPU. We report wall-clock time, memory required during the SVD computation, and the additional memory by SVD.

| Model | Unembedding Shape | Time | Peak Memory | Extra Memory |
|---|---|---|---|---|
| Qwen2.5-7B-Instruct | $152,064 \times 3,584$ | 1.30s | 8.37GB | 0.02GB |
| LLaMA-3.1-8B | $128,256 \times 4,096$ | 1.60s | 8.08GB | 0.03GB |
| Qwen2.5-72B-Instruct | $152,064 \times 8,192$ | 9.83s | 19.22GB | 0.13GB |
| Qwen3-235B-A22B-Instruct-2507-FP8 (MoE) | $151,936 \times 4,096$ | 1.70s | 9.55GB | 0.03GB |

**Space Complexity.** Storing the matrix $M$ requires

$$O(n^2 + nd + d^2)$$

memory. The truncated singular vectors $U \in \mathbb{R}^{n \times k}$ and $V \in \mathbb{R}^{d \times k}$ introduce an additional

$$O((n + d)k)$$

space overhead. Because $n$ is very large in modern LLMs, the memory is dominated by storing $U$.

**SVD Resource Consumption.** To quantify the practical resource requirements of performing SVD on the unembedding layer, Table 6 summarizes the wall-clock time, the peak memory consumption during the SVD computation, and the additional memory introduced by truncated SVD across several representative models. The evaluation covers models of different scales—including Qwen2.5-7B-Instruct, LLaMA-3.1-8B, Qwen2.5-72B-Instruct, and the MoE model Qwen3-235B-A22B-Instruct-2507-FP8 (Yang et al., 2025)—using an H100 80GB GPU.

**Singular Value Distribution in Larger Models.** Figure 12 shows the singular value distribution of the unembedding layers for larger models, including Qwen2.5-72B-Instruct and Qwen3-235B-A22B-Instruct-2507-FP8 (MoE). The trend of singular value decay is consistent with that observed for Qwen2.5-7B-Instruct and LLaMA-3.1-8B (Figure 4a), indicating that our method can be directly applied to larger models.

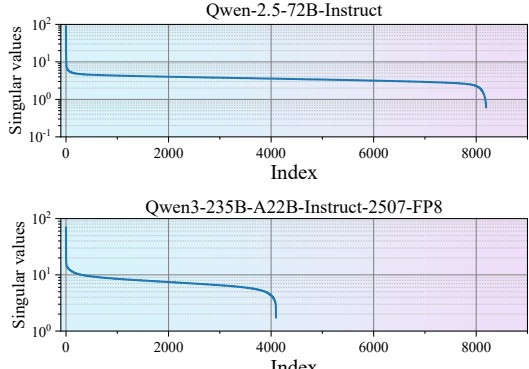

Figure 12: Singular value distributions of Wunemb after SVD, with hidden state dimensions of 8192 for Qwen2.5-72B-Instruct and 4096 for Qwen3-235B-A22B-Instruct-2507-FP8 (MoE).

