# OpenReview forum: "HARP: Hallucination Detection via Reasoning Subspace Projection"
_ICLR.cc/2026/Conference — ICLR 2026 Poster_

### Official Review · Reviewer_Vnht · 2025-10-31

**Soundness:** 1
**Presentation:** 2
**Contribution:** 2
**Rating:** 2
**Confidence:** 4

**Summary:**

This paper introduces HARP, a novel framework for detecting hallucinations in Large Language Models (LLMs). The central claim is that the final hidden state space of an LLM can be decomposed into a direct sum of two orthogonal subspaces: a "semantic subspace" and a "reasoning subspace." The authors posit that the model's unembedding layer parameter matrix, W_unemb, inherently separates these two spaces. By applying Singular Value Decomposition (SVD) to W_unemb, the authors identify the basis vectors for these subspaces. Specifically, right singular vectors corresponding to large singular values are proposed to span the semantic subspace, while those corresponding to near-zero singular values are claimed to span the reasoning subspace.
The proposed method, HARP, projects the final hidden states onto this identified "reasoning subspace" to generate low-dimensional features. These features are then fed into a two-layer MLP classifier to produce a token-level hallucination score. The final hallucination score for a given response is the maximum score across all its tokens. The authors conduct experiments on several open-domain QA datasets (NQ Open, TruthfulQA, TriviaQA, TyDiQA) using Qwen-2.5-7B and LLaMA-3.1-8B models, reporting state-of-the-art performance in terms of AUROC.

**Strengths:**

1. Novelty of Approach: The core idea of leveraging the spectral properties of the unembedding matrix to derive features for a downstream task is novel and interesting. It moves beyond standard probing or output-based consistency checks by attempting to ground the detection mechanism in the model's internal structure, which is a commendable direction for mechanistic interpretability.
2. Simplicity and Efficiency: The proposed method is technically straightforward and computationally efficient. It relies on a one-time SVD pre-computation and a simple projection, avoiding costly iterative sampling or complex architectural modifications. If validated and efficient at scale, this would represent a practical and scalable approach for hallucination detection.
3. Strong Empirical Results on Tested Benchmarks: The paper presents compelling empirical results, consistently outperforming several established baselines across multiple datasets and two different model architectures. The reported gains, such as a 7.5% AUROC improvement on TriviaQA, are significant and suggest that the extracted features are indeed highly informative for hallucination detection in the tested settings.

**Weaknesses:**

The paper suffers from fundamental weaknesses in its theoretical grounding and argumentation.
1. Unsupported Foundational Premise: The cornerstone of this work—the decomposition of the hidden state space into orthogonal "semantic" and "reasoning" subspaces—is presented as a discovery but is, in fact, an unsubstantiated assertion.
  ○ The paper transitions from a high-level cognitive analogy directly to a formal mathematical claim (Eq. 3) without providing any theoretical justification, empirical evidence from prior work, or logical deduction. Statements like "we establish that the hidden state space ... can be decomposed" are misleading; the property is not established, but is postulated.
  ○ The properties defined in Eq. 5 and Eq. 6 are the very definitions the authors introduce. The subsequent claim "Based on the properties..." is a form of circular reasoning. The validity of the entire framework rests on this unproven assumption, making the subsequent technical development an exercise built on a fragile foundation.
2. Misleading Terminology and Lack of Conceptual Rigor: The choice to label the near-null space of W_unemb as the "reasoning subspace" is highly speculative and lacks evidence.
  ○ The authors provide no causal or correlational evidence linking this subspace specifically to what is commonly understood as "reasoning" (e.g., chain-of-thoughs, self-reflection, etc.). An alternative, and perhaps more plausible, hypothesis is that this subspace captures any information orthogonal to the final token prediction, which could include contextual noise, computational artifacts, or simply redundant features.
  ○ Attaching the loaded term "reasoning" to this mathematically-derived space overstates the paper's findings and risks misdirecting future research. A more scientifically cautious term, such as "output-orthogonal subspace," would be more appropriate.
3. Arbitrary Formulation and Disconnected Sections: The mathematical modeling in Section 3.1, which defines hallucination via a similarity function sim and threshold λ, is both arbitrary and disconnected from the main technical contribution of the paper. This section does not inform the subsequent SVD-based methodology and its inclusion suggests a lack of a coherent, unified theoretical framework.
4. Limited Experimental Scope and Unaddressed Scalability Concerns: The empirical evaluation, while strong on its specific benchmarks, is narrow and fails to address critical questions about the method's applicability to larger, state-of-the-art models.
  ○ The experiments are confined to 7B/8B parameter models. It is unclear if the observed spectral properties of W_unemb and the effectiveness of HARP will generalize to much larger models (e.g., 70B+), where hallucination behaviors may differ.
  ○ The paper does not discuss the computational feasibility of performing a full SVD on the unembedding matrix of a model with a very large vocabulary and hidden dimension (e.g., >128k vocabulary, >8192 hidden dim). This is a critical practical consideration for the method's scalability.
  ○ Given the fact that Mixture-of-Experts (MOE) framework becomes status-quo choice in open-sourced and proprietary model developement, the experiment design does not reflect that its claim is applicable to a wide range of LLM design choices. The proposed method fails to generalize its impact in the mainstream research.

**Questions:**

To be more confident in my assessment, I would like the authors to address the following:
1. Can you provide any theoretical justification or cite prior work that supports the claim that the hidden state space naturally decomposes into orthogonal semantic and reasoning components, and that the unembedding layer acts as the separator? Beyond the analogy, what is the mechanistic basis for this hypothesis?
2. How do you defend the specific label "reasoning subspace"? Have you conducted any analysis to show that activity in this subspace correlates with explicit reasoning steps (e.g., in chain-of-thought prompting, mathematical problems, or logical puzzles) and is not just capturing other forms of non-semantic information?
3. Could you elaborate on the computational cost of applying SVD to the unembedding matrices of much larger models, such as those with vocabulary sizes exceeding 100,000 and hidden dimensions exceeding 8,192? Is the proposed method practical for state-of-the-art foundation models?
4. Why should the community believe that findings on 7B/8B models are representative of hallucination phenomena in significantly larger models? Have you considered any experiments or analysis to suggest this generalization holds?

---

> ### Author Response · Authors · 2025-11-21
>
> Thank you for the reviewer’s thoughtful and constructive questions. We have carefully revised the manuscript to address the concerns in detail. Our point-by-point responses are as follows:
>
> **Q1: Theoretical justification for orthogonal semantic and reasoning components.**
>
> We appreciate the request for more mechanistic clarification. The revised manuscript expands the discussion in **Section 4.2** (from Line 231 to the end of the section), formalizing how the interaction between the unembedding matrix (W_{\text{unemb}}) and a hidden vector (h) motivates a natural decomposition of feature directions aligned with vocabulary semantics versus complementary components that do not directly contribute to token prediction. This strengthened analysis supports the view that (W_{\text{unemb}}) acts as an effective separator between semantic and non-semantic components, providing theoretical justification beyond analogy.
>
> **Q2: Justification for the label “reasoning subspace.”**
>
> We appreciate the importance of demonstrating interpretability. Although constrained by space in the original version, we had already designed relevant experiments. We now include a **Reasoning Patch** experiment in **Appendix E**, based on mathematical reasoning tasks. The results show that patching reasoning components from correct solutions can effectively rectify erroneous reasoning trajectories while preserving semantic coherence, indicating a direct correlation between activity in this subspace and explicit reasoning processes. This supports the choice of terminology and demonstrates that the subspace is not merely capturing residual noise or unrelated hidden-state variations.
>
> **Q3: Computational cost and scalability of SVD.**
>
> We thank the reviewer for raising this practical consideration. In **Appendix F**, we provide additional analysis of computational efficiency and discuss the feasibility of applying SVD to larger models. We also extend our experiments to larger-scale architectures, including Qwen2.5-7B-Instruct and a MoE model, Qwen3-235B-A22B-Instruct. The results indicate that the observed behavior remains consistent and that the required SVD computation remains practical when amortized as a one-time preprocessing step.
>
> **Q4: Generalization beyond 7B/8B models.**
>
> To further support scalability expectations, the revised manuscript includes additional experiments on larger models, also detailed in **Appendix F**, showing similar phenomenology to the results reported in the main paper. Moreover, we note that recent hallucination detection research predominantly evaluates on mid-size LLMs, including **EigenScore (Chen et al., 2024)[1]**, **Haloscope (Du et al., 2024)[2]**, and **TSV (Park et al., 2025)[3]**. Consistent trends across these models suggest that phenomena identified at this scale are representative for studying hallucination mechanisms.
>
> We hope these revisions address the concerns and help clarify the broader applicability and theoretical grounding of our work. Thank you again for the insightful questions.
>
> [1] Chen, Chao, et al. "INSIDE: LLMs' internal states retain the power of hallucination detection." arXiv preprint arXiv:2402.03744 (2024).
>
> [2] Du, Xuefeng, Chaowei Xiao, and Sharon Li. "Haloscope: Harnessing unlabeled llm generations for hallucination detection." Advances in Neural Information Processing Systems 37 (2024): 102948-102972.
>
> [3] Park, Seongheon, et al. "Steer LLM Latents for Hallucination Detection." Forty-second International Conference on Machine Learning. (2025).

---

### Official Review · Reviewer_NZ6Q · 2025-10-31

**Soundness:** 3
**Presentation:** 3
**Contribution:** 2
**Rating:** 4
**Confidence:** 4

**Summary:**

This paper proposes HARP (Hallucination Detection via Reasoning Subspace Projection), a framework for detecting hallucinations in Large Language Models (LLMs).

**Strengths:**

1. Comprehensive Experimental Validation: The experiments cover diverse datasets (open-domain QA, reading comprehension) and models.

2. Novel Theoretical Insight: The paper introduces a meaningful decomposition of the LLM hidden state space into semantic and reasoning subspaces.

**Weaknesses:**

1. Limited Generalization to Larger Models: The experiments only use medium-sized LLMs (Qwen-2.5-7B-Instruct, LLaMA-3.1-8B). It remains unclear whether HARP can maintain its performance on larger-scale models (e.g., 70B or 175B parameters), as the structure of hidden state spaces and the role of the Unembedding layer may change with model scale.

2. Ambiguity in Reasoning Subspace Interpretation: While the paper defines the reasoning subspace as capturing internal reasoning processes, it lacks a more concrete, interpretable validation (e.g., linking specific components of the reasoning subspace to specific reasoning steps). This makes it difficult to fully confirm that the projected features truly reflect "reasoning" rather than other irrelevant hidden state information.

3. Sensitivity to SVD Parameter Selection: The paper sets the semantic subspace dimension k=d×95%, based on singular value distribution, but it does not thoroughly explore how different k values (e.g., 90%, 98%) affect detection performance across diverse datasets and models. This may limit the adaptability of HARP to different LLM architectures.

4. Lack of Comparison with Recent Advanced Methods: The baseline methods include relatively early works (e.g., Perplexity from 2023) but lack comparisons with the latest hallucination detection methods (post-2024) that also focus on model internal states. This makes it hard to fully position HARP’s competitiveness in the current state of the art.

5. Practical Deployment Challenges: Although HARP reduces feature dimensions, it still requires accessing the hidden states of LLMs—a requirement that may not be feasible for closed-source LLMs (e.g., GPT-4, Claude 3). The paper does not discuss solutions for such scenarios, limiting its practical application scope.

**Questions:**

See weaknesses.

---

> ### Author Response · Authors · 2025-11-21
>
> Thank you for the constructive and detailed feedback. We sincerely appreciate the reviewer’s thoughtful comments and have revised the manuscript accordingly. Our point-by-point responses are as follows:
>
> **W1: Limited Generalization to Larger Models.**
>
> We agree that examining larger-scale models is important for evaluating generalizability. In the revised version, we added new experiments in **Appendix F**, including both a larger dense model (e.g., Qwen2.5-7B-Instruct) and a Mixture-of-Experts architecture (e.g., Qwen3-235B-A22B-Instruct). The observed behaviors are consistent with the analysis in the main paper, further supporting the scalability of HARP across model sizes and architectures.
>
> **W2: Ambiguity in Reasoning Subspace Interpretation.**
>
> We appreciate the reviewer’s suggestion to clarify interpretability. Due to space limitations, the original submission did not include additional reasoning-based evidence. We now provide a **Reasoning Patch** experiment in **Appendix E**, conducted on mathematical reasoning datasets. The results demonstrate that patching reasoning components from correct solutions can effectively rectify erroneous reasoning trajectories while preserving semantic coherence, validating that the reasoning subspace indeed captures procedural reasoning information rather than unrelated hidden-state variations.
>
> **W3: Sensitivity to SVD Parameter Selection.**
>
> The original submission already discussed parameter sensitivity. As defined, the semantic and reasoning subspaces are orthogonal, and thus the reasoning dimension is (d-k). In **Figure 5(a)** we analyze semantic information loss as the reasoning dimension increases, and in **Figure 5(b)** we examine hallucination detection accuracy under varying reasoning dimensions. Based on this joint analysis, we find that (d-k=256) (~5%) yields a favorable trade-off, motivating our choice of (k = d \times 95%).
>
> **W4: Lack of Comparison with Recent Advanced Methods.**
>
> We acknowledge the importance of positioning HARP within the latest landscape. The initial version included contemporary comparisons such as **EigenScore (Chen et al., 2024)[1]** and **Haloscope (Du et al., 2024)[2]**, and we have further clarified and improved the presentation of this section in the revision.
>
> **W5: Practical Deployment Challenges.**
>
> We agree that access to hidden states is not feasible for closed-source models (e.g., GPT-4, Claude 3). However, we emphasize that this does not relate to our central contribution: demonstrating that hidden representations can be factorized into semantic and reasoning subspaces and that such decomposition enables effective hallucination detection without additional training. We discuss this limitation and potential future directions (e.g., distillation-based approximations) in the revised manuscript.
>
> We hope these revisions address the reviewer’s concerns and further strengthen the paper. Thank you again for your valuable feedback.
>
>
> [1] Chen, Chao, et al. "INSIDE: LLMs' internal states retain the power of hallucination detection." arXiv preprint arXiv:2402.03744 (2024).
>
> [2] Du, Xuefeng, Chaowei Xiao, and Sharon Li. "Haloscope: Harnessing unlabeled llm generations for hallucination detection." Advances in Neural Information Processing Systems 37 (2024): 102948-102972.

---

### Official Review · Reviewer_EBLo · 2025-11-01

**Soundness:** 2
**Presentation:** 2
**Contribution:** 2
**Rating:** 4
**Confidence:** 5

**Summary:**

This paper presents HARP (HAllucination detection via Reasoning subspace Projection), a framework for hallucination detection that mathematically decomposes the hidden state space into semantic and reasoning subspaces. The approach employs SVD on the Unembedding layer to derive basis vectors representing these subspaces, enabling hidden states to be projected onto the reasoning subspace, which then serves as the input to a hallucination detector. HARP achieves substantial feature compression (to about 5% of the original dimensionality) while isolating reasoning-related signals crucial for identifying hallucinations. Experiments across multiple datasets (NQ Open, TruthfulQA, TriviaQA, TyDiQA) and models show state-of-the-art AUROC performance, surpassing existing approaches and baselines.

**Strengths:**

1. Research Innovation: The paper offers a novel perspective by interpreting LLM hallucinations through the decomposition of the hidden state space into semantic and reasoning subspaces. This conceptual framing provides new insight into the underlying mechanisms of hallucination generation.
2. Computational Efficiency: By projecting hidden states onto the reasoning subspace, the method effectively compresses feature dimensionality to about 5% of the original space, significantly improving both computational efficiency and robustness.
3. Methodological Effectiveness: As shown in Table 1, HARP achieves substantially higher AUROC scores than competing baselines across four QA datasets and two LLMs. The results demonstrate strong consistency and robustness across different data domains and model architectures. The paper also conducts comprehensive ablation studies and analyses to support its conclusions.
4. Clarity of Presentation: The methodology is clearly described and easy to follow.

**Weaknesses:**

1. Overreliance on SVD and orthogonality assumptions. HARP relies heavily on the assumption that the hidden representations can be effectively decomposed via SVD into orthogonal semantic and reasoning subspaces. In practice, neural network weight matrices rarely yield strictly zero singular values, and the orthogonality assumption may not strictly hold. Although Section 4.3 mitigates this issue using an approximate low-rank solution, the reliability of this decomposition requires more rigorous empirical validation and theoretical justification.
2. Limited experimental scope and insufficient diversity. The current experiments are primarily conducted on QA datasets, which may not fully reflect the generalization ability of HARP. It would strengthen the paper to evaluate the method on a broader range of tasks—particularly those involving more complex logical reasoning (e.g., mathematical reasoning datasets) and non-QA settings. Moreover, experiments on additional model architectures and scales are necessary, since the compatibility of some methods may vary with model size and vocabulary distribution. Including additional evaluation metrics such as ECE or F1 would also provide richer insights. Finally, the paper does not clarify whether results are averaged over multiple runs or report variance, which raises concerns about the robustness and statistical significance of the findings.
3. Lack of analysis on overconfidence and hallucination dynamics. LLMs are known to exhibit overconfidence, particularly on datasets with high accuracy. It would be valuable to analyze how HARP behaves under such conditions—e.g., comparing hallucination patterns and performance changes across models and datasets with varying accuracy levels. A differential analysis between correct and incorrect responses could also offer deeper insight into how HARP mitigates hallucinations.

**Questions:**

See above.

---

> ### Author Response · Authors · 2025-11-21
>
> We sincerely thank the reviewer for their insightful and constructive feedback, which has helped us significantly improve our manuscript. Below, we provide a point-by-point response to the raised concerns, along with the corresponding revisions made in the updated version of our paper.
>
> **W1: Overreliance on SVD and Orthogonality Assumptions**
>
> We appreciate the reviewer's emphasis on the importance of validating the reliability of our subspace decomposition approach. In response, we have expanded our empirical and theoretical analysis to address this concern more rigorously.
>
> First, as discussed in Section 4.3 and illustrated in Figure 5(a) of the original manuscript, we quantitatively evaluated the impact of the approximate low-rank decomposition on semantic information preservation. Our results demonstrate that even at a reasoning subspace dimensionality of 256, the semantic loss remains acceptable—specifically, the original top-1 token consistently remains within the top-5 tokens after decomposition, with an average rank of top-3. This indicates that the decomposition effectively isolates reasoning components without critically compromising semantic integrity, thereby supporting the practical validity of our orthogonality assumptions.
>
> Furthermore, in the revised version, we have added Appendix F, which includes additional experiments on larger-scale models and Mixture-of-Experts (MoE) architectures. These results confirm that our decomposition approach remains robust across diverse model types, reinforcing the theoretical justification for using SVD-based methods in this context. We believe these additions provide a more comprehensive foundation for the reliability of our decomposition framework.
>
> **W2: Limited Experimental Scope and Insufficient Diversity**
>
> We thank the reviewer for highlighting the need for broader experimental validation. In the revised manuscript, we have substantially expanded our evaluation to address these points.
>
> First, we acknowledge that space limitations initially prevented us from including all relevant experiments. Following the reviewer's suggestion, we have now incorporated a "Reasoning Patch" experiment in the newly created Appendix E. This experiment, conducted on mathematical reasoning tasks (e.g., GSM8K), demonstrates that patching reasoning components from correct solutions can effectively rectify erroneous reasoning trajectories while preserving semantic coherence. This not only validates the generalization of HARP to complex, non-QA tasks but also confirms that the identified reasoning subspace captures procedural logic. The implementation code for this experiment has been included in the supplementary materials (see Reasoning_Patch.ipynb).
>
> Additionally, we have enriched our evaluation metrics by including F1 scores in the revised Figure 6(b), providing deeper insights into model performance beyond accuracy. Finally, to ensure robustness and statistical significance, all reported results are now based on the average of 10 independent runs. We hope these enhancements adequately address the reviewer's concerns regarding experimental diversity and reliability.
>
> **W3: Hallucination Patterns and Performance Across Accuracy Levels**
>
> Thank you for the valuable feedback. To address this point, we have expanded our analysis in the revised manuscript and added Table 2, titled “Distribution of known and unknown questions across four QA datasets.” Using the criterion in Equation 1, we label each question as Known when the model’s knowledge state contains the correct answer and Unknown when none of the 10 sampled responses include the correct answer. This distribution directly reflects that LLMs exhibit substantially different inherent answer accuracies across the four datasets, resulting in varying proportions of Known versus Unknown questions. By making these differences explicit, we are able to more precisely analyze how HARP behaves under heterogeneous accuracy conditions.
>
> We once again thank the reviewer for their valuable comments, which have greatly enhanced the quality of our work. We look forward to any further feedback and hope that the revisions adequately address all raised points.

---

### Official Review · Reviewer_VdiQ · 2025-11-01

**Soundness:** 3
**Presentation:** 3
**Contribution:** 2
**Rating:** 4
**Confidence:** 4

**Summary:**

This paper introduces HARP, a novel framework for hallucination detection in LLMs. The core hypothesis is that an LLM's hidden state space can be decomposed into a "semantic subspace" and a "reasoning subspace". The authors propose to identify these subspaces by applying SVD to the unembedding matrix. They define the "reasoning subspace" as the one spanned by the basis vectors corresponding to the smallest singular values. The method then projects the final-layer hidden states onto this low-dimensional "reasoning subspace" and uses these projections as features to train a simple MLP classifier for hallucination detection.

The paper reports state-of-the-art results on several QA datasets, suggesting high accuracy and efficiency.

**Strengths:**

The core idea of separating semantic and reasoning information by analyzing the unembedding matrix is novel and insightful.

The reported experimental results show a significant improvement over existing baselines, which is impressive.

**Weaknesses:**

The evaluation is confined to simplistic tasks that are misaligned with modern, critical LLM use cases.

The paper's entire empirical validation rests on short-form, factual question-answering datasets like NQ Open, TruthfulQA, and TriviaQA. While HARP demonstrates strong performance, these tasks are arguably "solved" problems for state-of-the-art LLMs and do not represent the true frontier of the current hallucination challenge. If this paper were submitted two years ago, its contribution would be significant, but the field has rapidly advanced. The critical applications where robust hallucination detection is most needed today are in:

- Long-form generation (e.g., summarization, report writing)

- Complex reasoning (e.g., multi-step math or logic problems)

- Agentic workflows (e.g., software development and tool use)

- ......

In these complex scenarios, hallucinations manifest as subtle logical fallacies, fabricated evidence, inconsistent reasoning steps, or non-existent API calls—not just the simple factual errors tested in this paper (e.g., "The capital... is Shanghai!").


The core assumption of HARP is that a "reasoning subspace" can be isolated and used for detection. However, the paper provides no evidence that this method can capture these more complex, structural, and procedural forms of hallucination. It is highly unclear whether the proposed technique can generalize beyond the simple, factual-recall domain to address the far more difficult and relevant problems facing LLMs today.

----

The paper appears to be missing a critical and directly relevant baseline: Unsupervised Real-Time Hallucination Detection based on the Internal States of Large Language Models.

This work is highly relevant as it proposes a method for hallucination detection that also leverages the model's internal states. Specifically, the authors of that paper develop a Classifier that receives the contextualized embeddings of each token during the LLM’s inference process. It then outputs a probability indicating the probability of hallucination in the LLM’s output.

Given the strong methodological overlap (i.e., using internal representations/embeddings during inference for classification), this paper should be considered a key baseline.

I strongly recommend that the authors:

- Discuss this paper in the related work section.

- Include it as a baseline in the experimental comparison and discuss how the current submission's performance and methodology differ.




I am open to increasing my score if the authors address these points.

**Questions:**

Please refer to the previous section

---

> ### Author Response · Authors · 2025-11-21
>
> We sincerely thank the reviewers for their insightful and constructive comments, which have provided us with an excellent opportunity to clarify and strengthen the presentation of our work. We have revised the manuscript accordingly and provide a point-by-point response below.
>
> **W1: Evaluation on Complex Tasks**
>
> We thank the reviewer for this critical point. We fully agree that validating hallucination detection methods on complex tasks like long-form generation and reasoning is crucial, as hallucinations in these scenarios manifest as structural and logical inconsistencies far beyond simple factual errors.
>
> We would like to clarify that, aligned with this vision, we had in fact already designed and conducted experiments to probe this aspect prior to the review. Specifically, we performed a "Reasoning Patch" experiment on mathematical reasoning tasks (e.g., GSM8K) to test whether the identified "reasoning subspace" can capture procedural logic. The results show that patching internal representations from correct solutions can rectify erroneous reasoning trajectories.
>
> This analysis was initially omitted from the manuscript due to space constraints. Following the reviewer's insightful suggestion, we have now incorporated these findings into the newly created Appendix E of the revised manuscript. Furthermore, the implementation code for the Reasoning Patch experiment has been included in the updated supplementary materials (see Reasoning_Patch.ipynb). We hope this addition directly addresses the reviewer's concern by providing empirical evidence that HARP generalizes effectively to the complex, multi-step tasks that define the current frontier of hallucination challenges.
>
> **W2: Discussion of Relevant Baseline**
>
> We thank the reviewer for raising the important point regarding the baseline of unsupervised hallucination detection using internal states. We appreciate this opportunity to clarify our original contributions.
>
> In our initially submitted manuscript, we had already integrated this line of research both in discussion and empirical comparison. Specifically:
>
> 1. The Related Work section contained a discussion of contemporary methods that leverage internal states for hallucination detection, explicitly citing and discussing EigenScore (Chen et al., 2024) [1] and Haloscope (Du et al., 2024) [2].
>
> 2. The experimental results (presented in Table 1) included a direct performance comparison with these methods, demonstrating HARP's competitive advantages.
>
> Upon carefully considering the reviewer's comment, we recognized that the presentation of this comparative analysis, while present, could be structured more clearly to immediately orient the reader. In response, we have refactored the relevant passages in the Related Work and Experiments sections.
>
> Once again, we are grateful for the thorough and engaging review, which has been instrumental in enhancing the clarity and impact of our manuscript.
>
> [1] Chen, Chao, et al. "INSIDE: LLMs' internal states retain the power of hallucination detection." arXiv preprint arXiv:2402.03744 (2024).
>
> [2] Du, Xuefeng, Chaowei Xiao, and Sharon Li. "Haloscope: Harnessing unlabeled llm generations for hallucination detection." Advances in Neural Information Processing Systems 37 (2024): 102948-102972.

---

> ### Comment · Reviewer_VdiQ · 2025-11-27
> **Thanks for the reply!**
>
> Thanks for the reply!
>
> Your reply address my concern, so I increase my score and revise my review accordingly.

---

> > ### Author Response · Authors · 2025-11-28
> >
> > Thank you for your thoughtful consideration and for revising your assessment.
> >
> > We sincerely appreciate your time and constructive feedback throughout the review process.

---

### Meta-Review · Area_Chair_kYPy · 2026-01-08

**Summary:**

This paper presents a framework for hallucination detection, HARP, which mathematically decomposes the unembedding matrix into semantic and reasoning subspaces using SVD. Beyond  feature compression, HARP is able to isolate reasoning-related signals crucial for identifying hallucinations. Experiments across multiple QA datasets and models show state-of-the-art performance of HARP.

Overall, I found that all reviewers’ concerns about the papers were very addressable, and the authors did a good job of addressing each through their rebuttal as well as additional experimentation / analysis.

**Reviewer Concerns:**

**Addressed**
1. Authors introduce experiments on more complex tasks - addressing the comments by Vdiq, eblo.
2. Authors include empirical comparisons with relevant baselines from recent work.
3. Authors address eblo’s concerns about Overreliance on SVD and Orthogonality Assumptions by including experiments on more complex architectures.
4. Authors address eblo’s concerns about Hallucination Patterns and Performance Across Accuracy Levels through an additional analysis of the results.
5. Assumption that hidden states can be decomposed into semantic and reasoning subspaces (vdiq, eblo, vnht), which the authors have tried to address through argumentation and experimentation. However, such assumptions are hard to justify fully. In my opinion, this could be considered an assumption of the work, and all the results in the paper could rest on this assumption, which is not unreasonable, given the positive empirical results.

**Unaddressed**
None that I could determine.

**Reviewer Scores:**

- Vdiq responds in text stating that they increased their scores as the authors successfully addressed adding a complex task and comparing with relevant baselines.
- All other (eblo, vnht, nz6q) reviewer’s concerns are to the best of my understanding adequately addressed in the response by the authors, and I would assume that they would increase their scores, if they had responded.

---

### Decision · Program_Chairs · 2026-01-26

Accept (Poster)